# Recreational Physical Activity and the Mediterranean Diet: Their Effects on Obesity-Related Body Image Dissatisfaction and Eating Disorders

**DOI:** 10.3390/healthcare12161579

**Published:** 2024-08-08

**Authors:** Ioannis Tsartsapakis, Panagiotis Papadopoulos, Dionisis Stavrousis, Athanasios A. Dalamitros, Athanasia Chatzipanteli, Georgios Chalatzoglidis, Maria Gerou, Aglaia Zafeiroudi

**Affiliations:** 1Department of Physical Education and Sport Science, Aristotle University of Thessaloniki, 62122 Serres, Greece; papadoppv@phed-sr.auth.gr (P.P.); dionisisstavrousis@gmail.com (D.S.); gchalatzo@phed-sr.auth.gr (G.C.); miamixan@hotmail.com (M.G.); 2Department of Physical Education and Sport Science, Aristotle University of Thessaloniki, 54124 Thessaloniki, Greece; dalammi@phed.auth.gr; 3Department of Physical Education and Sport Science, University of Thessaly, 42100 Trikala, Greece; atchatzip@yahoo.gr (A.C.); azafeiroudi@uth.gr (A.Z.)

**Keywords:** Mediterranean diet, lifestyle, obesity, exercise, health promotion, body image

## Abstract

Obesity is a non-communicable disease that is associated with a number of serious physical and mental health conditions. The present study examines the effect of recreational physical activity and the Mediterranean diet on body image dissatisfaction and propensity for eating disorders. It is based on 1311 participants categorized by body mass index (BMI) into a normal ΒΜΙ group (NBG; N = 513), an overweight and obese ΒΜΙ group (OBG; N = 492), and a control group (CG; N = 309). All participants completed the Multidimensional Body-Self Relations Questionnaire-Appearance Scale (MBSRQ-AS), Rosenberg Self-esteem Scale (RSES), Eating Attitudes Test-26 (EAT-26), Mediterranean Diet Score (MedDietScore), and Fitness Evaluation and Fitness Orientation subscales from the original Multidimensional Body-Self Relations Questionnaire (MBSRQ). The results of the multiple regression analysis indicated that the overall prediction of the variables was statistically significant. A multivariate analysis of variance (MANOVA) demonstrated the existence of significant interactions between groups and gender across a range of scales. Despite higher body image dissatisfaction in the OBG group, they maintained positive self-esteem and did not exhibit eating disorder tendencies. Notably, women reported greater dissatisfaction than men across all three groups. Our findings have practical implications for public health promotion strategies, policymaking, future research, and clinical practice. Encouraging regular exercise and adherence to the Mediterranean diet could improve body satisfaction and reduce eating disorder risk. Policymakers can advocate for community-based policies promoting physical activity and healthy eating habits.

## 1. Introduction

Recreational physical activity, organized exercise, and dietary habits play a pivotal role in maintaining human health [1,2]. They significantly influence the musculoskeletal system, organ health, mental well-being, and overall life quality [3,4,5]. Obesity, characterized by excessive fat accumulation leading to a body mass index (BMI) greater than 30 kg/m^2^, is a chronic, complex disease with significant implications for health [6]. The prevalence of obesity can be attributed to a multitude of factors, including a sedentary lifestyle, physical inactivity, unhealthy dietary habits, and genetic predisposition [7,8]. In addition to the rise in obesity, these factors contribute to the simultaneous development of non-communicable diseases such as metabolic syndrome, type 2 diabetes, and cardiovascular disease [9,10]. According to the World Health Organization, these diseases are responsible for 70% of total mortality [11].

The Mediterranean diet, which has been the subject of numerous studies demonstrating its protective effects against obesity, cardiovascular disease, and diabetes [12,13], emphasizes the consumption of seasonal, fresh, minimally processed, traditional, local, and environmentally friendly products. This includes vegetables, fruits, extra virgin olive oil, wholegrain breads and cereals, moderate consumption of dairy products, fish, and fish products, thereby maximizing the intake of protective nutrients and promoting sustainability [14]. This dietary pattern is in contrast with Western-style consumption habits, which include convenience eating, the consumption of pre-packaged foods, the consumption of refined grains, the consumption of processed red meat, the consumption of high-sugar drinks, the practice of smoking, and the consumption of alcohol in excess [15].

The Mediterranean lifestyle, which views nutrition as a social opportunity and promotes the passing of healthy food habits to future generations, integrates movement as an essential component [16]. Both the Mediterranean diet and recreational physical activity are inversely associated with obesity [17,18] and positively associated with a healthy ΒΜΙ and normal weight [19]. Regular physical activity helps burn calories, maintain muscle mass, and promote fat loss [20], and it is associated with elevated self-esteem, positive body image, and lower levels of tension, anxiety, and depression [21].

Besides the well-documented adverse effects of obesity on physical health, recent focus has been directed towards its association with mental health [22]. Obesity has been linked to various psychiatric disorders, including personality disorders, chronic mood disorders, eating disorders, schizophrenia, and attention-deficit/hyperactivity disorder [23]. It can also result from the side effects of medications prescribed for mental health conditions [23]. Another issue related to obesity is body image dissatisfaction, which is a significant mental-health concern [24,25]. Body dissatisfaction is defined as a personal dislike of one’s body or specific body parts [26,27]. This dissatisfaction can give rise to a range of strong emotional, perceptual, and psychological reactions, as it is reflected as a dynamic state [28]. Consequently, body dissatisfaction represents one of the principal characteristics and risk factors associated with individuals diagnosed with eating disorders [29,30].

Moreover, body dissatisfaction can contribute to chronic diseases in non-clinical populations, including unsuccessful smoking cessation, reduced sexual functioning, lower physical-health-related quality of life, and inadequate attention to cancer screening [31,32]. Furthermore, research suggests that women experience higher levels of body dissatisfaction than men. This discrepancy may arise from women’s tendency to place greater importance on their physical appearance throughout life [33,34]. In addition, research indicates that body dissatisfaction can reduce the motivation to engage in physical activity, strive for weight loss, and maintain a healthy diet [5,35]. However, it is important to note that not all obese individuals are equally susceptible to these conditions, and individuals of normal weight may also experience significant body dissatisfaction [24].

Despite extensive evidence on the impact of all components of the Mediterranean lifestyle on human health, comprehensive studies assessing the impact of the Mediterranean lifestyle on body dissatisfaction and eating disorders are scarce [15,36,37]. Eating disorders (EDs) represent complex psychiatric illnesses, characterized by excessive preoccupation with food, unhealthy weight control measures, and dissatisfaction with body image [38]. Although there is a substantial body of evidence concerning the influence of various elements of the Mediterranean lifestyle on overall health, there is a paucity of comprehensive studies that have specifically examined the impact of this lifestyle on body dissatisfaction and eating disorders. Consequently, the objective of the present study is to address this gap in the literature by investigating the relationship between recreational physical activity, adherence to the Mediterranean diet, and their influence on body dissatisfaction and eating disorder tendencies. The present study explores these associations across individuals with varying degrees of BMI, with the hypothesis that differences exist among these groups.

## 2. Materials and Methods

### 2.1. Study Design and Procedure

The present study is a cross-sectional investigation. The questionnaires were distributed by researchers between November 2022 and March 2024. The researchers visited sports and recreation centers (leisure and sports parks, centers, and gyms) or made appointments with groups organizing recreational physical activities and distributed the questionnaires after providing appropriate explanations for completing them. The control subjects were recruited from a variety of locations, including work offices, discussion groups, and department stores. Incomplete questionnaires were excluded from the survey. Participants who reported pregnancy, using dietary supplements, or taking weight-loss medication were also excluded from the study.

To investigate adherence to the Mediterranean diet, the questionnaire from Panagiotakos et al. [39] was used. Recreational physical activity was defined as an exercise during leisure time performed for fitness, relaxation, or pleasure. The data on participation in the exercise were derived from the questionnaire of demographic characteristics, which was prepared for the purposes of the study. In order to be eligible for the study, participants in the ‘normal BMI’ group (NBG) and ‘overweight and obese BMI’ group (OBG) were required to engage in recreational physical activity and exercise for a minimum of four days per week for a duration of at least 45 min (running, cycling, trail running, cross-training, spinning, treadmill running, dance aerobics, or any other form of exercise deemed appropriate by the researchers). The exercise must have been performed for a period of at least two years [40]. Participants in the control group (CG) were required to have not exercised regularly in the last two years. The Multidimensional Body-Self Relations Questionnaire-Appearance Scale (MBSRQ-AS) questionnaire was employed to assess the levels of body image dissatisfaction exhibited by the participants. Furthermore, the Eating Attitudes Test-26 (EAT-26) questionnaire was used to assess the propensity for eating disorders, while the Rosenberg Self-esteem Scale (RSES) questionnaire assessed participants’ overall self-esteem. The study was approved by the Internal Ethics Committee (IEC) of the Department of Physical Education and Sport Science, Aristotle University of Thessaloniki, Greece (ERC-11/2024), and complies with the Declaration of Helsinki.

### 2.2. Participants 

A convenience sampling method was performed in order to recruit the participants subsequently involved in the study. The entire cohort of participants was comprised of individuals hailing from Greece. The study included 1311 participants aged 34.4 ± 9.99 years (18–59 years old), of whom 693 (52.9%) were men and 618 (47.1%) were women. The participants were divided into three groups based on their BMI and exercise habits. The NBG consisted of 513 individuals 33.6 ± 9.56 years of age, of whom 236 (46%) were men and 277 (54%) were women. The OBG included 492 overweight or obese individuals 36.8 ± 9.90 years of age, of whom 365 (74.2%) were men and 127 (25.8%) were women. The CG comprised 306 individuals 32.1 ± 10.1 years of age with an average normal BMI, of whom 92 (30.1%) were men and 214 (69.9%) were women. All participants voluntarily joined the study after being informed about its purpose and signing a participation form.

### 2.3. Instrumentation

(A)Anthropometric and demographic characteristics. This questionnaire was created for the study and included items such as gender, age, weight, and height. The participants’ BMI was calculated based on their self-reported weight and height using the formula BMI = weight (kg)/height (m)^2^. Additionally, participants provided information about how many years they had exercised, the type(s) of exercise they participated in, the duration of their exercise routine, and the frequency of their exercise routine. Finally, participants were asked a closed-ended question to state the main reason that motivated them to start exercising.(B)RSES. The RSES [41,42] is a reliable psychometric tool for measuring overall self-esteem. The questionnaire consists of 10 items. The questionnaire is based on the unidimensional model of the self-concept and measures global self-esteem through 10 questions, five of which are positive and five of which are negative. Questions include “I feel that I am a person of worth, at least on an equal plane with others” and “I feel I do not have much to be proud of”. Answers were given on a four-point Likert-type scale ranging from ‘strongly agree’ to ‘strongly disagree’. The questionnaire is intended for individuals aged 12 and above. The scale has also been used in the Greek population with statistically significant results, validity, and reliability [43,44,45]. The author suggests that higher scores indicate higher levels of self-esteem.(C)EAT-26. The EAT-26 [46] is a self-report questionnaire consisting of 26 questions that address attitudes, beliefs, and concerns about food, body shape, and weight. Respondents answer on a 6-point Likert-type scale ranging from 1 (never) to 6 (always). Items 1–25 are rated on a 4-point scale (0 = sometimes, rarely, or never; 1 = often; 2 = usually; 3 = always), while item 26 is reverse-scored. The final score is calculated by summing items 1–26. Examinees who score 20 or higher belong to the risk group for eating disorders. It is worth noting that the EAT-26 is not a diagnostic instrument but rather an indicator of tendencies towards eating disorders. The scale’s validity has been documented in numerous international and Greek studies [47,48].(D)Mediterranean Diet Score (MedDietScore): The MedDietScore is a questionnaire [39] that assesses adherence to the Mediterranean diet using a composite dietary index comprising 11 components. These components represent the main food groups of the Mediterranean diet, including unprocessed cereals, fruits, vegetables, potatoes, legumes, olive oil, fish, red meat, poultry, whole dairy products, and alcohol. The scoring scale is extensive, enhancing its predictive accuracy. The theoretical range of the questionnaire spans from 0 to 55. Adherence to the Mediterranean diet is categorized based on the MedDietScore as follows: low compliance (0 < MedDietScore ≤ 20), moderate compliance (21 < MedDietScore ≤ 35), or high compliance (36 < MedDietScore ≤ 55). The validity of the MedDietScore for the Greek population has been established in previous studies [49,50].(E)MBSRQ-AS: The MBSRQ-AS [51,52], a condensed version of the MBSRQ-69, comprises 34 items across five subscales: Appearance Evaluation, Appearance Orientation, Body Areas Satisfaction Scale (BASS), Overweight Preoccupation, and Self-classified Weight. Each subscale is scored individually by summing the points allocated to its items and dividing by the total number of items in the subscale.The ‘Appearance Evaluation’ subscale assesses an individual’s appraisal of their appearance and satisfaction therewith. Those who score highly on the aforementioned scale tend to feel positive and satisfied with their appearance. Conversely, those who score low on the scale tend to exhibit a general unhappiness with their physical appearance.The ‘Appearance Orientation’ subscale gauges the psychological importance of appearance to an individual and the degree to which their cognitions and behaviors are oriented around their physical appearance Those who achieve high scores tend to place a greater emphasis on their physical appearance, paying closer attention to their grooming habits and engaging in extensive self-grooming. Those with low scores are indifferent to their appearance; their appearance is not a significant concern, and they do not expend a great deal of effort to “look good”.The ‘Body Areas Satisfaction Scale’ measures satisfaction or dissatisfaction with specific body parts. Individuals who achieve high composite scores are typically content with the majority of their body. Those with low scores are dissatisfied with the size or appearance of multiple areas.The ‘Overweight Preoccupation’ subscale evaluates the degree of concern about weight control, including dietary restrictions. Those with high composite scores demonstrate a tendency towards body image concerns, weight monitoring, dieting, and eating restraint.The ‘Self-Classified Weight’ subscale assesses an individual’s perception of their weight, spanning from underweight to overweight. Those who achieve the highest scores on this subscale demonstrate a dissatisfaction with their body weight and have an estimated weight that is higher than their actual weight.The scale’s validity has been documented in numerous international and Greek studies [53].(F)MBSRQ Fitness Evaluation and Fitness Orientation [51]. The Fitness Evaluation subscale of MBSRQ measures an individual’s perception of their physical condition and level of involvement in physical activities. A high score indicates a high level of fitness and engagement in sports. The MBSRQ Fitness Orientation subscale measures an individual’s investment in their fitness. A high score indicates that the individual places importance on being fit and regularly participates in activities that enhance or maintain their fitness. The validity of the MBSRQ for the Greek population has been established in previous studies [53].

### 2.4. Data Analysis

Statistical analyses were conducted using IBM SPSS Statistics ver. 29.0 (IBM Co., Ltd., Armonk, NY, USA). The normal distribution of the data was evaluated using the Kolmogorov–Smirnov test, which showed that the obtained samples were normally distributed. To measure the internal consistency of the scales, Cronbach’s Alpha was used (refer to Section 3.1.3). Descriptive analysis was conducted on physical characteristics such as age, height, weight, and ΒΜΙ to determine the means and standard deviations. The study analyzed exercise factors, including years of experience, exercise duration and frequency per week, and initial motivation for physical activity. A multivariate analysis of variance (MANOVA) was conducted to examine the differences among the three groups and gender in terms of body dissatisfaction, self-esteem, eating attitudes, adherence to the Mediterranean diet, and orientation and evaluation of fitness status. Partial eta-squared (η*_p_*^2^) values were also computed to indicate the effect size. Pearson correlation analysis was conducted to test the correlation between the variables of body dissatisfaction, BMI, eating attitudes, Mediterranean diet, self-esteem, fitness orientation, and fitness evaluation. Regression analysis was then performed to predict the variables’ impact on body dissatisfaction and eating attitudes. The statistical significance threshold was set at *p* < 0.05.

## 3. Results

### 3.1. Descriptive Statistics

The results of the descriptive statistical analysis, including the mean and standard deviation for all variables in the survey questionnaire, can be found in the Appendix A.

#### 3.1.1. Anthropometric Characteristics of the Participants

Table 1 presents the anthropometric characteristics of participants and for the NBG, OBG, and CG as means and standard deviations for the variables age (years), weight (kg), height (m), and BMI (kg/m^2^). A statistically significant difference was observed between the three groups in terms of weight (*F*_(2.1308)_ = 360.029, *p* = 0.001). In particular, the mean body weights of the NBG and CG were found to be significantly lower than that of the OBG (*p* < 0.001 and *p* < 0.001, respectively). Furthermore, a statistically significant difference was observed between the three groups in terms of BMI (*F*_(2.1308)_ = 726.899, *p* = 0.001). In particular, the NBG exhibited a significantly lower BMI than the OBG and CG (*p* < 0.001 and *p* < 0.001, respectively). Furthermore, the CG exhibited a significantly lower BMI than the OBG group (*p* < 0.001).

#### 3.1.2. Years of Exercise 

The NBG had exercised regularly for 4.8 ± 2.1 years. The majority (88.7%) exercised four to five times a week for a maximum of two hours. The OBG had participated in regular recreational (see Section 2.1) exercise for 3.3 ± 1.6 years. The majority of participants (77.9%) reported exercising regularly, between four and five times per week, for a maximum of two hours.

#### 3.1.3. Reliability Analysis

To assess the internal consistency of the scales, we calculated Cronbach’s α coefficient. The EAT-26 scale had a Cronbach’s α coefficient of 0.85, the RSES scale had a Cronbach’s α coefficient of 0.84, and the MedDietScore scale had a Cronbach’s α coefficient of 0.91. The Cronbach’s α coefficients for the MBSRQ-AS Appearance Evaluation, Appearance Orientation, BASS, Overweight Preoccupation, and Self-classified Weight subscales were 0.84, 0.72, 0.82, 0.71, and 0.86, respectively. For the MBSRQ fitness evaluation and fitness orientation subscales, the Cronbach’s α coefficients were 0.73 and 0.83, respectively. These values are considered satisfactory.

### 3.2. Pearson Correlation Analyses 

Correlation analysis was conducted to examine the relationships between MedDietScore, self-esteem, BMI, fitness evaluation, and fitness orientation with the subscales of MBSRQ-AS and EAT-26. The results of the correlation analysis are given in Table 2.

### 3.3. Multiple Regression Analyses

Five stepwise multiple regression analyses were conducted for all participants, with dependent variables including MedDietScore, Self-Esteem, BMI, EAT-26, Fitness Evaluation, and Fitness Orientation. The independent variables were the subscales of MBSRQ-AS: Appearance Evaluation, Appearance Orientation, BASS, Overweight Preoccupation, and Self-Classified Weight. An additional stepwise multiple regression analysis was conducted with the EAT-26 as the dependent variable and the MedDietScore, Self-Esteem, BMI, Fitness Evaluation, Fitness Orientation, and five subscales of the MBSRQ-AS as independent variables, as shown in Table 3.

### 3.4. Multivariate Analysis of Variance 

A MANOVA with three groups and gender as factors revealed significant interactions in MBSRQ-AS subscales: Appearance Evaluation (*F*_(2.1306)_ = 25.858, *p* < 0.001, η*_p_*^2^ = 0.038), BASS (*F*(_2.1306)_ = 11.331, *p* < 0.001, η*_p_*^2^ = 0.017), Overweight Preoccupation (*F*_(2.1306)_ = 12.017, *p* < 0.001, η*_p_*^2^ = 0.018), Self-Classified Weight (*F*_(2.1306)_ = 40.656, *p* <0.001, η*_p_*^2^ = 0.059), EAT-26 (*F*_(2.1306)_ = 3.781, *p* = 0.023, η*_p_*^2^ = 0.006), and RSES (*F*_(2.1306)_ = 6.718, *p* < 0.001, η*_p_*^2^ = 0.010). Despite gender differences, consulting norms for each gender in MBSRQ-AS subscales is recommended [51]. Post hoc analyses were conducted separately for each gender.

The Bonferroni post hoc test revealed significant differences in the Appearance Evaluation subscale and consequently in physical appearance satisfaction among the three groups. Men in the NBG were significantly more satisfied with their appearance than those in the OBG (*p* < 0.001) and CG (*p* < 0.001). The OBG exhibited a tendency towards dissatisfaction. Women in the NBG also showed a positive evaluation of their appearance, contrasting with the negative evaluation in the OBG, while women in the CG had a positive view (Figure 1).

No significant differences were found among men in the Appearance Orientation subscale (*p* > 0.05). However, overweight and obese women (OBG) were more appearance-oriented compared with those in the NBG (*p* = 0.008) and CG (Figure 1).

The MBSRQ-AS BASS subscale results showed significant differences. Men in the NBG were more satisfied with their body areas than those in the OBG (*p* < 0.001) and CG (*p* = 0.004). For women, the NBG showed higher satisfaction than the OBG (*p* < 0.001) and CG (*p* = 0.011). Women in the OBG had marginal satisfaction with their body areas (Figure 1).

The MBSRQ-AS Overweight Preoccupation subscale showed that men in the NBG were significantly less preoccupied with being overweight than those in the OBG (*p* < 0.001) and CG (*p* < 0.001). Women in the OBG showed significantly higher anxiety about their body fat and weight than those in the NBG (*p* < 0.001) and CG (*p* < 0.001) (Figure 1).

The MBSRQ-AS Self-Classified Weight subscale showed that the NBG had a significantly different weight perception compared with the OBG (*p* < 0.001) and CG (*p* < 0.001). Men in the NBG group perceived their weight as lower, while those in the OBG and CG groups perceived it as higher. Women in the OBG group perceived their weight as higher than actual, unlike those in the NBG (*p* < 0.001) and CG (*p* < 0.001) groups, who perceived it as normal or lower (Figure 1).

The MBSRQ Fitness Evaluation subscale showed significant differences between the NBG and OBG (*p* = 0.018) and CG (*p* = 0.010) for men, and between the NBG and OBG (*p* < 0.001) and CG (*p* < 0.001) for women. 

The MBSRQ Fitness Orientation subscale revealed significant differences between the NBG and OBG (*p* = 0.003) and CG (*p* < 0.001) for men, and between the NBG and OBG (*p* < 0.001) for women. Both men and women in the NBG valued fitness and were actively engaged in fitness activities. 

The EAT-26 questionnaire showed significant differences in eating disorder propensity among the three groups (*F*_(2.1309)_ = 77.032, *p* < 0.001, η*_p_*^2^ = 0.106). The NBG differed significantly from the OBG (*p* < 0.001) and CG (*p* < 0.001), and the CG differed from the OBG (*p* < 0.001). No group or gender had a tendency towards the development of eating disorders, as their mean scores on the EAT-26 questionnaire were below twenty (<20).

The RSES questionnaire revealed significant self-esteem differences among the groups (*F*_(2.1309)_ = 111.415, *p* < 0.001, η*_p_*^2^ = 0.146). The NBG had significantly higher self-esteem than the OBG (*p* < 0.001) and CG (*p* < 0.001).

The MedDietScore questionnaire showed significant differences in adherence to the Mediterranean diet among the groups (*F*_(2.1309)_ = 1149.389, *p* < 0.001, η*_p_*^2^ = 0.638). The NBG demonstrated significantly higher adherence compared to the OBG (*p* < 0.001) and CG (*p* < 0.001). The OBG also differed significantly from the CG (*p* < 0.001). The NBG showed high compliance, while the OBG and CG showed moderate compliance with the diet.

## 4. Discussion

### 4.1. Observations and Findings

The findings of the present study provide valuable insights into the complex interplay between recreational physical activity, dietary habits, body dissatisfaction, self-esteem, and the propensity for eating disorders among individuals across different BMI groups. The results substantiate the hypothesis that there would be differences in body dissatisfaction and propensity for eating disorders between individuals of the three groups.

The MANOVA results revealed that the NBG exhibited the most positive body image for both genders. In contrast, the OBG exhibited the highest levels of body dissatisfaction, as indicated by their MBSRQ-AS scores. This was true for both genders. The participants in the OBG exhibited the highest BMI compared with the other groups. The results indicate a significant negative association between BMI and the Appearance Evaluation, Appearance Orientation, and BASS subscales of the MBSRQ-AS, as well as with the MedDietScore, Self-esteem, Fitness Evaluation, and Fitness Orientation subscales of the MBSRQ. Conversely, BMI was found to be significantly positively associated with the Overweight Preoccupation and Self-classified Weight subscales of the MBSRQ-AS, as well as with the EAT-26. The results of the present study corroborate those of previous studies, which indicate that a high BMI value is directly related to body dissatisfaction and the occurrence of eating disorders in both women and men [54,55].

It is noteworthy that individuals in the OBG who engage in regular recreational physical activity and adhere to the Mediterranean diet, despite exhibiting higher body dissatisfaction compared with other groups, still maintain a positive degree of self-esteem and do not show tendencies to develop eating disorders. This indicates a complex relationship between self-esteem, the Mediterranean diet, recreational exercise, BMI, body dissatisfaction, and the propensity for eating disorders among overweight and obese individuals [56,57,58]. The above relationship has been demonstrated in several studies, including that by Brechan and Kvalen [59], who concluded that the effect of body dissatisfaction on disordered eating is mediated through self-esteem and depression. Similarly, Lampard et al. [60] concluded that self-esteem mediates the relationship between interpersonal problems and eating disorder symptoms. Body dissatisfaction has both direct and indirect effects on disordered eating through self-esteem and negative affect [61].

The findings of our study indicate that self-esteem is significantly positively associated with the MBSRQ-AS Appearance Evaluation and BASS subscales, as well as MedDietScore and the MBSRQ Fitness Evaluation and Fitness Orientation subscales. Subsequently, a significantly negative association between self-esteem and the MBSRQ-AS subscales Appearance Orientation, Overweight Preoccupation, and Self-Classified Weight; BMI; and the EAT-26. It is crucial to highlight that this correlation is negative. Individuals with higher self-esteem tend to exhibit lower levels of body dissatisfaction and disordered eating, as evidenced by several studies [61,62,63].

The results were corroborated by regression analysis, which demonstrated that self-esteem is a significant positive predictor for the Appearance Evaluation and BASS subscales, and it is a significant negative predictor for the propensity to develop eating disorders (EAT-26). These findings are also consistent with and confirm previous studies in which a negative relationship between self-esteem and eating disorders is shown [64,65].

Adherence to the Mediterranean diet is inversely correlated with the propensity to manifest eating disorders [66,67]. The moderate adherence to the Mediterranean diet observed in the OBG may be indicative of a positive change in their physical condition due to regular recreational physical activity [68,69,70]. The review by Obeid et al. [71] indicates that, over the past decade, the Mediterranean population has exhibited moderate adherence to the Mediterranean lifestyle. This adherence is observed across all age groups and genders within the Mediterranean population [71]. A period of global economic recession, spanning from 2007 to 2010, had a significant impact on adherence to the Mediterranean diet owing to the influence of socioeconomic factors [72]. The findings of the aforementioned studies provide some support for the moderate adherence to the Mediterranean diet observed among the OBG and CG participants. Additionally, both men and women in the OBG demonstrate a strong orientation towards achieving their desired physical fitness, as evidenced by their significantly higher scores on the Fitness Orientation subscale when compared with the CG [73]. The results indicate that the Mediterranean diet and regular recreational exercise play a mediating role in the absence of eating disorder tendencies and high self-esteem demonstrated by the OBG participants.

In the present study, the Mediterranean diet (MedDietScore) was significantly positively associated with the MBSRQ-AS Appearance Evaluation and BASS subscales and significantly negatively associated with the MBSRQ-AS Overweight Preoccupation and Self-Classified Weight subscales as well as being significantly negatively associated with EAT-26. The results of the present study are in agreement with those of Godoy-Izquierdo et al. [74], who concluded that adherence to the Mediterranean dietary pattern is associated with a number of beneficial outcomes, including better mental health and psychological positivity, as well as increased subjective well-being, among overweight or obese Spanish adults.

The participants in the NBG and CG exhibit normal BMI values within the healthy range. However, multivariate analysis of variance revealed that participants in the NBG exhibit more positive body image, higher self-esteem, higher adherence to the Mediterranean diet, and a lower tendency to develop eating disorders compared with the CG for both genders. The NBG was engaged in regular recreational physical activity , with high adherence to the Mediterranean diet. In contrast, the CG did not engage in regular recreational physical activity , with moderate adherence to the Mediterranean diet. These findings are consistent with those of Martinovits et al. [68], who posit that individuals who engage in more physical activity may be more cognizant of the significance of a nutritious and balanced diet for their well-being. A number of studies have demonstrated that the adoption and maintenance of a healthy diet and regular exercise are crucial for a good health-related quality of life [69,70,75,76].

It is noteworthy that there is a significant positive correlation between the Fitness Evaluation and Fitness Orientation variables, which assess the subjective perception of physical fitness or unfitness of participants, and the MedDietScore score in the NBG. There are no discernible gender differences. The results of this study are in agreement with those of Bizzozero-Peroni et al. [77] and Zurita-Ortega et al. [69], who have demonstrated that high adherence to the Mediterranean diet is associated with higher physical fitness in adults. This is consistent with the findings observed for the NBG. Moreover, several other studies show the benefits of high adherence to the Mediterranean diet [78,79]. The Mediterranean diet, characterized by high consumption of fruits, vegetables, legumes, and healthy fats, has been associated with numerous health benefits, including weight management and improved psychological well-being. This nutritional model helps to improve body composition and favors a decrease in the percentage of fat mass [80]. The present study found that adherence to the Mediterranean diet was significantly higher in the NBG compared with the OBG and CG. This is significantly correlated with higher self-esteem, lower body dissatisfaction, higher physical fitness, and a reduced tendency towards eating disorders. This indicates that the Mediterranean diet, when combined with regular physical activity, may confer a beneficial effect on mental health and body image.

One of the notable findings of this study is the higher levels of body dissatisfaction reported by women compared with men. The findings of this study align with those of previous research, which similarly demonstrated that women exhibit a higher prevalence of body dissatisfaction than men [81,82,83]. These results are more significant in the OBG than in the other two groups. This gender disparity in the OBG is consistent with the existing literature, which indicates that women generally place greater importance on their physical appearance and are more likely to experience body dissatisfaction [84,85]. This heightened body dissatisfaction among women in the OBG could be attributed to societal pressures and cultural norms that emphasize thinness and ideal body shapes for women [86,87]. Interestingly, despite higher body dissatisfaction, women in the OBG maintain a positive degree of self-esteem and do not show a heightened propensity for developing eating disorders. This paradoxical finding suggests that while body dissatisfaction is prevalent, it does not necessarily translate to lower self-esteem or disordered eating behaviors, possibly due to the mediating effects of regular exercise and a healthy diet.

The results of this study indicate that encouraging adherence to the Mediterranean diet in conjunction with recreational exercise guidelines may offer a more comprehensive approach to achieving enhanced health benefits, in addition to those derived from the Mediterranean diet and exercise alone. The socializing and communicative effects of physical activities and recreational exercise create a multitude of positive effects on different categories of practitioners [88].

### 4.2. Limitations of the Study

Nevertheless, it is crucial to acknowledge the limitations inherent in this study. Primarily, the reliance on self-reported data may introduce bias, as participants may over- or underestimate their adherence to the Mediterranean diet, their exercise habits, or their levels of body satisfaction or dissatisfaction.

Secondly, the cross-sectional design of the study precludes the establishment of causality. Longitudinal studies would be required to ascertain cause-and-effect relationships.

Thirdly, the study’s narrow demographics could also be considered a limitation. The study’s participants are primarily from one specific geographic location (Greece), and the results may not be generalizable to other populations. Furthermore, body dissatisfaction was estimated considering factors such as self-esteem, eating disorders, regular exercise, and adherence to the Mediterranean diet. However, a number of factors may influence the relationships between the aforementioned factors and body dissatisfaction. These include psychological factors, sociocultural factors, environmental factors, genetic factors, personal preferences and habits, health status, and significant others (family, friends, and community resources).

Finally, it should be noted that the regression analysis did not take into account a number of factors that have been demonstrated to significantly affect the results of such surveys, including place of residence, economic status, and social status.

### 4.3. Implications

The findings of this study have several implications for public health promotion strategies, policymaking, future research, and clinical practice.

Firstly, encouraging regular recreational exercise and adherence to the Mediterranean diet could potentially improve body satisfaction and reduce the risk of eating disorders.

Secondly, the findings could be used to advocate for policies that promote physical activity and healthy eating habits in the community.

Thirdly, our study could facilitate further research into the intricate relationships between diet, exercise, body dissatisfaction, self-esteem, and eating disorders. Future studies could examine these relationships in different populations or employ longitudinal designs to investigate cause-and-effect relationships.

Fourthly, the findings of our study could be beneficial in clinical practice. For instance, clinicians working with individuals who exhibit high levels of body dissatisfaction or who are at risk for eating disorders may wish to consider the potential benefits of regular recreational exercise and adherence to the Mediterranean diet.

Obesity is associated with significant morbidity, both physically and mentally. In order to prevent and reduce the occurrence of these problems, the authors propose the following strategies: firstly, participation in individual or group recreational exercise programs on a regular basis without participating in competitions that add stress and compulsive exercise; secondly, following a healthy diet that will enhance and accentuate the benefits of recreational exercise; thirdly, it is recommended that individuals form friendships with others who share similar interests and focus less on physical appearance and more on the health benefits and enjoyment derived from these relationships.

## 5. Conclusions

The findings suggest that promoting a combination of recreational physical activity and adherence to the Mediterranean diet can offer comprehensive health benefits beyond those achieved by either intervention alone. This approach not only enhances physical health but also supports psychological well-being and social interaction, contributing to a higher quality of life. In practical terms, health promotion strategies should emphasize regular recreational exercise and the Mediterranean diet to improve body satisfaction and reduce eating disorder risks. Specifically, overweight and obese individuals who regularly engage in recreational physical activity and adhere to the Mediterranean diet maintain higher self-esteem and do not exhibit tendencies toward eating disorders, despite experiencing greater body dissatisfaction.

## Figures and Tables

**Figure 1 healthcare-12-01579-f001:**
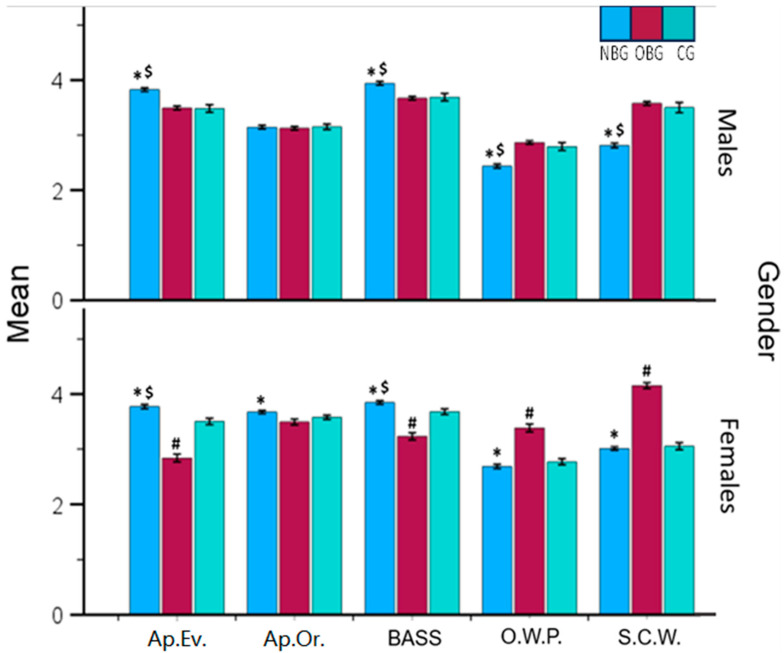
Comparison of MBSRQ-AS scores between the three groups, stratified by gender. NBG = blue bar, OBG = red bar, CG = green bar. Ap.Ev. = Appearance Evaluation, Ap.Or. = Appearance Orientation, BASS = Body Areas Satisfaction Scale, O.W.P. = Overweight Preoccupation, S.C.W. = Self-Classified Weight. ***** = significant difference between NBG and OBG. $ = significant difference between NBG and CG. # = significant difference between OBG and CG.

**Table 1 healthcare-12-01579-t001:** Anthropometric characteristics of the participants.

		Age (Years)	Height (m)	Weight (kg)	BMI (kg/m^2^)
	N	M ± SD	M ± SD	M ± SD	M ± SD
Total Sample	1311	34.4 ± 9.99	1.74 ± 0.080	73.8 ± 14.6	24.9 ± 3.71
NBG	513	33.6 ± 9.56	1.74 ± 0.080	67.1 ± 9.91	22.0 ± 1.84
OBG	492	36.8 ± 9.90	1.76 ± 0.076	85.1 ± 10.1	28.2 ± 2.26
CG	306	32.1 ± 10.1	1.71 ± 0.079	66.9 ± 16.2	24.6 ± 3.75

NBG = normal weight BMI group, OBG = overweight and obesity BMI group, CG = control group.

**Table 2 healthcare-12-01579-t002:** Pearson Correlation Analyses.

	Ap.Ev.	Ap.Or.	BASS	Ov.W.Pr.	S.C.W.	EAT-26	MDS.	RSES	BMI	Fit.Ev.	Fit.Or.
Ap.Ev.	1										
Ap.Or.	0.044	1									
BASS	0.656 **	0.019	1								
O.W.P.	−0.378 **	0.281 **	−0.327 **	1							
S.C.W.	−0.463 **	0.022	−0.439 **	0.437 **	1						
EAT-26	−0.437 **	0.189 **	−0.296 **	0.494 **	0.247 **	1					
MDS.	0.241 **	0.020	0.208 **	−0.191 **	−0.239 **	−0.280 **	1				
RSES	0.455 **	−0.099 **	0.397 **	−0.317 **	−0.226 **	−0.528 **	0.377 **	1			
BMI	−0.334 **	−0.203 **	−0.244 **	0.289 **	0.552 **	0.318 **	−0.377 **	−0.237 **	1		
Fit.Ev.	0.406 **	−0.011	0.439 **	−0.123 **	−0.230 **	−0.004	0.139 **	0.219 **	−0.038	1	
Fit.Or.	0.302 **	0.047	0.291 **	−0.031	−0.253 **	0.072 **	0.268 **	0.116 **	−0.059 *	0.535 **	1

Ap.Ev. = Appearance Evaluation, Ap.Or. = Appearance Orientation, BASS = Body Areas Satisfaction Scale, O.W.P. = Overweight Preoccupation, S.C.W. = Self-Classification Weight, EAT-26 = Eating Attitudes Test-26, MDS. = MedDietScore, RSES = Rosenberg Self-Esteem Scale, BMI = Body Mass Index, Fit.Ev. = Fitness Evaluation, Fit.Or. = Fitness Orientation. ** = Correlation is significant at the 0.01 level (2-tailed), * = Correlation is significant at the 0.05 level (2-tailed).

**Table 3 healthcare-12-01579-t003:** Multiple regression analyses.

MBSRQ-AS Subscale	*R* ^2^	Adj.*R*^2^	Predictors	Coefficient Beta	95% CI
Appearance Evaluation	0.426	0.425	Self-Esteem	0.182	[0.025, 0.044]
			Fitness Evaluation	0.377	[0.341, 0.429]
			BMI	−0.163	[−0.041, −0.024]
			EAT-26	−0.271	[−0.031, −0.021]
Appearance Orientation	0.112	0.111	EAT-26	0.282	[0.018, 0.026]
			BMI	−0.292	[−0.057, −0.039]
BASS	0.336	0.334	Fitness Evaluation	0.424	[0.343, 0.426]
			Self-Esteem	0.206	[0.026, 0.044]
			BMI	−0.167	[−0.019, −0.010]
Overweight Preoccupation	0.277	0.275	BMI	0.134	[0.017, 0.036]
			EAT-26	0.446	[0.037, 0.046]
			Fitness Evaluation	−0.118	[−0.162, −0.070]
Self-Classified Weight	0.368	0.367	BMI	0.527	[0.104, 0.123]
			Fitness Orientation	−0.144	[−0.248, −0.118]
			Fitness Evaluation	−0.147	[−0.218, −0.105]
**EAT-26**	0.451	0.449	Overweight Preoccupation	0.318	[2.946, 3.897]
			Fitness Orientation	0.197	[1.613, 2.576]
			Self-Esteem	−0.370	[−0.823, −0.641]
			Appearance Evaluation	−0.240	[−3.040, −1.975]

Adj. *R*^2^ = Adjusted *R*^2^, BASS = Body Areas Satisfaction Scale, EAT-26 = Eating Attitudes Test-26.

## Data Availability

The data presented in this study are available on request from the corresponding author.

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
