# Peer review of "Recreational Physical Activity and the Mediterranean Diet: Their Effects on Obesity-Related Body Image Dissatisfaction and Eating Disorders"

_healthcare, 2024, doi:10.3390/healthcare12161579_

Round 1
Reviewer 1 Report
Comments and Suggestions for Authors
The study entitled, “Recreational physical activity and the Mediterranean diet: Their effects on obesity-related body-image dissatisfaction and eating disorders” encompasses original findings on the effects of recreational physical activities and the use of the Mediterranean diet on body-image dissatisfaction and eating disorders. The following changes are recommended in the manuscript:
1. The keywords should be revised by replacing the existing keywords with single-word, concise, and highly relevant keywords.
2. The mean values presented in Table 1 need to be compared statistically.
3. Data in Table 2 shall be properly described in textual form in Section 3.2. “Pearson Correlation Analyses”.
4. Data in Table 3 shall be elaborated in Section 3.3. “Multiple Regression Analysis”. The Coefficient Beta and their association with the MBSRQ-AS Subscale parameters need to be elaborated in sufficient detail.
5. Authors shall explain whether the study participants were selected randomly, or a non-random approach was adopted. Randomization is important to minimize bias in the study. Moreover, the data was based on self-reports of the participants through questionnaires, so it is even more important to explain how the bias was reduced by the authors.
6. In Table 2, the significance of the values of the correlation coefficient needs to be confirmed statistically.
Author Response
Authors Answers to 1st reviewer
Thank you for giving us the opportunity to submit a revised draft of the manuscript. We appreciate the time and effort that you dedicated to providing feedback on our manuscript and are grateful for the insightful comments on and valuable improvements to our paper. We have incorporated most of your suggestions. Those changes are highlighted within the manuscript. Please see below, in red, for a point-by-point response to your comments and concerns.
- The keywords should be revised by replacing the existing keywords with single-word, concise, and highly relevant keywords.
Answer: We are grateful for your proposal. We replace most of the keywords with single-word one.
- The mean values presented in Table 1 need to be compared statistically.
Answer: Thank you for your suggestion. In terms of BMI and therefore weight and height they have differences. After all, this was an important point of the research design. We believe that the differences in height and age between the groups, even though they exist, are not a factor that could significantly affect the processing and results of the research. The analysis of all these results would add an additional large text to the paper but without any contribution to the explanation of the results.
- Data in Table 2 shall be properly described in textual form in Section 3.2. “Pearson Correlation Analyses”.
Answer: Thank you. Describing the results in detail would excessively increase the size of the text. We believe that table 2 clearly shows the correlations between the research variables. In the discussion part, special mention is made of the correlations between the variables where the relationship between them is highlighted. In the discussion chapter, particular attention is drawn to the correlations between the variables, with a particular focus on the relationships between them (see lines 381-388, 392-393, 401-406, 409-413, 415-416, 431-439, 452-455, 465-468).
- Data in Table 3 shall be elaborated in Section 3.3. “Multiple Regression Analysis”. The Coefficient Beta and their association with the MBSRQ-AS Subscale parameters need to be elaborated in sufficient detail.
Answer: Thank you for pointing this out. However, another reviewer asked to delete the detailed explanation of the results and present them in a table. We could bring back the analytical explanations of Multiple Regression Analysis. That would contradict the other reviewer. However, we list below the results as originally written: “Five stepwise multiple regression analyses were conducted for all participants, with independent variables including MedDietScore, Self-Esteem, BMI, EAT-26, Fitness Evaluation, and Fitness Orientation. The dependent variables were the subscales of MBSRQ-AS: Appearance Evaluation, Appearance Orientation, BASS, Overweight Pre-occupation, and Self-Classified Weight. An additional stepwise multiple regression analysis was conducted with the EAT-26 as the dependent variable and the MedDietScore, Self-Esteem, BMI, Fitness Evaluation, Fitness Orientation, and the five sub-scales of the MBSRQ-AS as independent variables.
The overall prediction of the variables was statistically significant for the entire sample (R2 = .426, Adjusted R2 = .425, F(4,1307) = 242.432, p < .001). The MBSRQ-AS “Appearance Evaluation” subscale was significantly positively predicted by Self-Esteem (.182, 95% CI [.025, .044]) and Fitness Evaluation (.377, [.341, .429]), and negatively correlated with BMI (-.163, [-.041,-.024]) and EAT-26 (-.271, [-.031, .021]).
The MBSRQ-AS “Appearance Orientation” subscale exhibited a significant overall prediction of the variables (R2 = .112, Adjusted R2 = .111, F(2,1305) = 82.422, p < .001), with a positive prediction of the EAT-26 (.282, [.018, .026]) and a negative association with BMI (-.292, [-.057, -.039]).
The MBSRQ “BASS” subscale demonstrated a significant overall predictive capacity for the variables (R2 = .336, Adjusted R2 = .334, F(3,1307)=220.054, p < .001), with a positive prediction of the Fitness Evaluation (.424, [.343, .426]) and the Self-Esteem (.206, [.026, .044]), and a negative correlation with BMI (-.167, [-.019, -.010]).
The MBSRQ-AS “Overweight Preoccupation” subscale demonstrated a significant overall predictive capacity for the variables (R2 = .277, Adjusted R2 = .275, F(3,1308) =166.826, p < .001), with a positive association by BMI (.134, [.017, .036]) and EAT-26 (.446, [.037, .046]), and a negative correlation with Fitness Evaluation (-.118, [-.162, -.070]).
The MBSRQ-AS “Self-Classified Weight” subscale demonstrated a significant overall predictive capacity for the variables (R2 = .368, Adjusted R2 = .367, F(3,1308)=253.615, p < .001), with a positive prediction of BMI (.527, [.104, .123]), a negative association of Fitness Orientation (-.144, [-.248, -.118]) and a negative correlation with Fitness Evaluation (-.147, [-.218, -.105]).
The EAT-26 demonstrated a significant overall predictive capacity for the variables (R2 = .451, Adjusted R2 = .449, F(4,1303)=266.826, p < .001), with a positive prediction by Overweight Preoccupation (.318, [2.946, 3.897]) and Fitness Evaluation (.197, [1.613, 2.576]), and a negative association with Self-Esteem (-.370, [-.823, -.641]) and Appearance Evaluation (-.240, [-3.040, -1.975]), (Supplementary S1).”
- Authors shall explain whether the study participants were selected randomly, or a non-random approach was adopted. Randomization is important to minimize bias in the study. Moreover, the data was based on self-reports of the participants through questionnaires, so it is even more important to explain how the bias was reduced by the authors.
Answer: In paragraph 2.1. "Study design and procedure" and in paragraph 2.2. "Participants", the participants are a convenience sample and the questionnaires were given to them live, hand to hand, over a period of about a year and a half. Thus, the researchers had the opportunity to check and notice if any of the participants declared false information. Furthermore, the sampling was completely random as none of the researchers knew in advance the participants and which of them would agree to participate in the research.
- In Table 2, the significance of the values of the correlation coefficient needs to be confirmed statistically.
Answer: We apologize but we don’t understand what you mean by saying that the significance of correlation coefficient values ​​should be confirmed statistically. We believe that the appropriate statistical analysis was performed, and the results presented are exactly as given by the particular statistical analysis. For your convenience, we attach the results of the analysis below for you to check.

Reviewer 2 Report
Comments and Suggestions for Authors
This study reported the effect of recreational physical activity and the Mediterranean die on obesity-related body-image dissatisfaction and eating disorders through a cross-sectional questionnaire investigation. Although some interesting findings have been revealed, there are still significant issues that need to be addressed.
(1) 3.2 Table 2 showed the results of pearson correlation analyses. What are the implications or conclusions that can be drawn from these analyses? There is no description.
(2) 3.3. The results of multiple regression analyses were shown in Table 3. What are the conclusions that can be drawn from these analyses? The authors have not addressed.
(3) There were 306 participants in the control group, which were required to have not exercised regularly in a longer two years. How to ensure that this was done or ensure the data was accurate?
(4) Minor concerns.
Figure 1, the names of three groups are suggested to be shown in the figure, not in the figure legend. This is easy for readers to understand the figure.
Table 2, Column 1, “Apso.” should be “Ap. Or.”
Table 1, row 2 “N M ±SD M ± SD M ± SD M ± SD” can be deleted.
Table 3. Column 1. The symbol “” can be deleted.
Author Response
Authors Answers to 2nd reviewer
Thank you for giving us the opportunity to submit a revised draft of the manuscript. We appreciate the time and effort that you dedicated to providing feedback on our manuscript and are grateful for the insightful comments on and valuable improvements to our paper. We have incorporated most of your suggestions. Those changes are highlighted within the manuscript. Please see below, in red, for a point-by-point response to your comments and concerns.
(1) 3.2 Table 2 showed the results of pearson correlation analyses. What are the implications or conclusions that can be drawn from these analyses? There is no description.
Answer: Thank you for your suggestion. According to the other reviewers, this paper is already quite long and we reduced its length by limiting the detailed reports for each analysis. However, you will find many references to the correlations in the discussion section where they are extensively presented and the significance of these correlations is explained according to the results (L381-388, L392-393, L401-406, L409-413, L415-416, L431-439 , L452-455, L465-468).
(2) 3.3. The results of multiple regression analyses were shown in Table 3. What are the conclusions that can be drawn from these analyses? The authors have not addressed.
Answer: Thank you for pointing this out. The results of the multiple regression analysis as well as the pearson correlation analysis were incorporated into the discussion to reduce the pages of the paper which were much more as you rightly observed. So, we decided to give more ground to the discussion and limit the results. However, for your convenience we list the results of the multiple regression analyzes below:
“Five stepwise multiple regression analyses were conducted for all participants, with independent variables including MedDietScore, Self-Esteem, BMI, EAT-26, Fitness Evaluation, and Fitness Orientation. The dependent variables were the subscales of MBSRQ-AS: Appearance Evaluation, Appearance Orientation, BASS, Overweight Pre-occupation, and Self-Classified Weight. An additional stepwise multiple regression analysis was conducted with the EAT-26 as the dependent variable and the MedDietScore, Self-Esteem, BMI, Fitness Evaluation, Fitness Orientation, and the five sub-scales of the MBSRQ-AS as independent variables.
The overall prediction of the variables was statistically significant for the entire sample (R2 = .426, Adjusted R2 = .425, F(4,1307) = 242.432, p < .001). The MBSRQ-AS “Appearance Evaluation” subscale was significantly positively predicted by Self-Esteem (.182, 95% CI [.025, .044]) and Fitness Evaluation (.377, [.341, .429]), and negatively correlated with BMI (-.163, [-.041,-.024]) and EAT-26 (-.271, [-.031, .021]).
The MBSRQ-AS “Appearance Orientation” subscale exhibited a significant overall prediction of the variables (R2 = .112, Adjusted R2 = .111, F(2,1305) = 82.422, p < .001), with a positive prediction of the EAT-26 (.282, [.018, .026]) and a negative association with BMI (-.292, [-.057, -.039]).
The MBSRQ “BASS” subscale demonstrated a significant overall predictive capacity for the variables (R2 = .336, Adjusted R2 = .334, F(3,1307)=220.054, p < .001), with a positive prediction of the Fitness Evaluation (.424, [.343, .426]) and the Self-Esteem (.206, [.026, .044]), and a negative correlation with BMI (-.167, [-.019, -.010]).
The MBSRQ-AS “Overweight Preoccupation” subscale demonstrated a significant overall predictive capacity for the variables (R2 = .277, Adjusted R2 = .275, F(3,1308) =166.826, p < .001), with a positive association by BMI (.134, [.017, .036]) and EAT-26 (.446, [.037, .046]), and a negative correlation with Fitness Evaluation (-.118, [-.162, -.070]).
The MBSRQ-AS “Self-Classified Weight” subscale demonstrated a significant overall predictive capacity for the variables (R2 = .368, Adjusted R2 = .367, F(3,1308)=253.615, p < .001), with a positive prediction of BMI (.527, [.104, .123]), a negative association of Fitness Orientation (-.144, [-.248, -.118]) and a negative correlation with Fitness Evaluation (-.147, [-.218, -.105]).
The EAT-26 demonstrated a significant overall predictive capacity for the variables (R2 = .451, Adjusted R2 = .449, F(4,1303)=266.826, p < .001), with a positive prediction by Overweight Preoccupation (.318, [2.946, 3.897]) and Fitness Evaluation (.197, [1.613, 2.576]), and a negative association with Self-Esteem (-.370, [-.823, -.641]) and Appearance Evaluation (-.240, [-3.040, -1.975]), (Supplementary S1).”
(3) There were 306 participants in the control group, which were required to have not exercised regularly in a longer two years. How to ensure that this was done or ensure the data was accurate?
Answer: Thank you for your question. We have accordingly revised to emphasize this point. According to chapter 2.1. "Study design and procedure", the completion of the questionnaires was done voluntarily. The researchers visited the participants' workplaces and asked first which of those who wanted to participate had not exercised for at least the last two years. Participants received absolutely no payment for completing the questionnaire and therefore had no reason to lie. However, we have included the issue you rightly mention in the research limitations (L491-494).
(4) Minor concerns.
Figure 1, the names of three groups are suggested to be shown in the figure, not in the figure legend. This is easy for readers to understand the figure.
Answer: You are absolutely right about this point. We added it to Figure 1.
Table 2, Column 1, “Apso.” should be “Ap. Or.”
Answer: We have changed it according to your suggestions.
Table 1, row 2 “N M ±SD M ± SD M ± SD M ± SD” can be deleted.
Answer: We appreciate the reviewer's insightful suggestion and agree. However, after a related search in similar papers in the journal, we notice that these were included in the tables and thus decided not to delete. If you consider it very important we can proceed with deletion.
Table 3. Column 1. The symbol “” can be deleted.
Answer: We have changed it according to your suggestions (L322-332).

Reviewer 3 Report
Comments and Suggestions for Authors
The paper offers an insightful analysis of how lifestyle interventions can influence physical and mental health outcomes. Although the research is timely and relevant, some major concerns should be considered. The most important one is the lack of novelty and the need for a clear explanation of contribution.
Author Response
Answers to 3rd reviewer
Thank you for giving us the opportunity to submit a revised draft of the manuscript. We appreciate the time and effort that you dedicated to providing feedback on our manuscript and are grateful for the insightful comments on and valuable improvements to our paper. We have incorporated most of your suggestions. Those changes are highlighted within the manuscript. Please see below, in red, for a point-by-point response to your comments and concerns.
The paper offers an insightful analysis of how lifestyle interventions can influence physical and mental health outcomes. Although the research is timely and relevant, some major concerns should be considered. The most important one is the lack of novelty and the need for a clear explanation of contribution.
Authors response: Thank you for your suggestions. After the analytical review of the literature, it emerged that several studies dealt with the benefits of the Mediterranean diet for health and weight loss. Similar results emerge for the benefits of exercise. However, we observed that there is very little research that has addressed the effect of the Mediterranean diet and exercise (ie Mediterranean lifestyle) on the psychological health of both normal weight and overweight and obese people. The major issue is that the economic crisis that hit Europe and especially Greece from 2009 to 2016 caused many people to turn to a convenient diet based on cheap ready-to-eat/standardized/pre-made products with the consequence that obesity rates in our country and the consequences it brings to people's health. We certainly didn't invent anything new. However, we are trying through this research to redefine where we are regarding the effort of many people and states for a return to healthy eating and physical activity which was essentially a common feature of the peoples of the Mediterranean. As physical activity in the form of work has long been lost in western-type societies (automation of work), exercise and indeed recreational exercise is the one that comes to fill the void left by physical manual work. This research aims to return us to the basic values ​​of the Mediterranean way of life that for several centuries have sustained us and given us the health (physical and mental) that we all seek. The contribution of the present research is therefore that it gives the clear message that if we return to the ancestral Mediterranean way of life, confirming previous researches. So, we have a lot of hope to improve the quality of life, psychological health and human health in general.

Reviewer 4 Report
Comments and Suggestions for Authors
Thank you for submitting to Healthcare. Please refer to the attached document.

Author Response
Authors Response to 4th reviewer
Thank you for giving us the opportunity to submit a revised draft of the manuscript. We appreciate the time and effort that you dedicated to providing feedback on our manuscript and are grateful for the insightful comments on and valuable improvements to our paper. We have incorporated most of your suggestions. Those changes are highlighted within the manuscript. Please see below, in red, for a point-by-point response to your comments and concerns.
abstract
There is no need to indicate ‘ ‘ in the group name.
The title has physical activity. There is no research related to PA in the research method. And the results talk about image satisfaction, but do not show the results of physical activity. Nevertheless, there is something about 'regular exercise' in the conclusion.
In the results, use clear variable names rather than 'overall' or 'multiple scales'.
Answer: Thank you for your suggestion. We agree to remove the ' ' from the group names. The title refers to Recreational physical activity. In line 109 we explain in detail what we mean by this term: Recreational physical activity was defined as an exercise during leisure time performed for fitness, relaxation, or pleasure. We apologize but we do not understand what you mean when you refer to the term "image satisfaction". This term is not in DSM-IV or DSM-V, so we can't be sure what to answer. In the results, if we add the names of all the variables we used in this research then the text size will increase dramatically. It has already been requested by another reviewer to reduce the length of the text.
Introduction
The overall content flow of the introduction must be revised. Due to the structure of the study, the introduction should begin with content about obesity, not physical activity. It is recommended that it be written in the following order.
* Content related to obesity
* Characteristics of obese people (lifestyle characteristics: diet, physical activity), (psychological characteristics: image dissatisfaction, eating disorders)
* A review of research on the relationship between simple lifestyle characteristics and psychological characteristics of previous studies.
(e.g., people who are obese but are physically active have higher self-esteem)
* Limitations of previous studies, essential purpose of this study, need for research on new discoveries
Answer: Thank you very much for your valuable suggestions. However, according to the title of the research, recreational physical activity is mentioned first and is a key innovation in the present research and thus we believe that it was correctly placed first in the introduction. Obesity is analyzed quite meticulously according to previous studies as well as the reviews of other reviewers. According to your suggestions for the introduction part no body image changes are suggested. So we think you enjoyed our report on body dissatisfaction. You have also not suggested anything about the Mediterranean diet and the Mediterranean lifestyle, and therefore we assume that you agree with the analysis of these terms as well. Thank you very much.
Research Method: This is the most important part of the study and the author's most flawed part.
According to the author's title, the design should be as follows.
So, it is correct to compare the image dissatisfaction and eating disorders of the groups.
In addition, there are many studies related to image dissatisfaction in obese people and have been conducted for a long time. I don't think it's a new design or concept.
Answer: Thank you for suggesting a better way to organize our research. However, we apologize, but the table you provide us is not completely understandable to us: More specifically, we ask you to explain the following so that we can properly proceed with the changes you request:
1) You divided the participants into two groups a) Mediterranean Diet, b) No Mediterranean Diet. In the present research there are three groups a) NBG, b) OBG and c) CG. You also have a group that does not follow the Mediterranean diet. We have three groups that all follow the Mediterranean diet.
2) You divide the two groups (there are three as we mentioned above) into High PA and Low PA. This does not refer to the purpose of the present research and we consider that it does not add anything essential to the results. As authors, the purpose of the research was to have the sample participate in recreational physical activity, and we consider it to be something innovative in research. We don't care about performance and we didn't deal with athletes and contestants. Also according to the separation, you made it is not clear to us what to do with the control group. They don't exercise at all.
In the results, gender differences are one of the main results. It should be added to Table 1.
And obesity is very complex and there is a lot of information we have to consider simultaneously. To do so, authors must provide as much general, demographic, or sociological information as possible.
Answer: Thank you for pointing this out. The results for gender differences are in the supplementary table we have attached. As you will notice these results are at least a page and a half and would increase the text size dramatically. So we preferred to give them as a supplementary folder. After all, it has already been requested to reduce the length of the text by at least half.
Added additional Table 2. Information on intergroup scores on various questionnaires should be provided. (e.g. mean, standard deviation, 95%CI, effect size, median, range). It is not right to only show the results of whether there is a relationship between variables or not without this information.
Answer: The information you request is in the supplementary file.
There are no restrictions on references. However, there seem to be too many unnecessary references. For example, lines 39-40 “They significantly influence the musculoskeletal system, organ health, mental well-being, and overall life quality” are references 4, 5, 6, and 7, even though they are not terribly new information. One or two of these are enough. We hope to review it as a whole and delete many references. There are 48 references in the introduction alone. It is thought that it can be reduced to less than half.
Answer: We appreciate your feedback however authors prefer to leave the bibliography references as they are as they consider them necessary to more thoroughly support those mentioned in the paper. After all, as you mention, there is no limit on reports. Thank you very much.

Reviewer 5 Report
Comments and Suggestions for Authors
Dear Authors,
It is a very carefully write article with interesting results and useful insight for practical applications.
We have only small corrections/ suggestions for your consideration.
Line 88 to 95, the study is well justified, you write about 3 groups but you already did not define them, maybe it is better to reserve that for material and methods, because it is not very relevant for the introduction.
In Materials and methods, we know in which kind of institutions the subjects were recruited, but we do not know in witch condition they responded to the questionnaires (online? Other?), please specify.
Table 1, please correct the 176 for Height in OBG
Author Response
Authors Response to 5th reviewer
Thank you for giving us the opportunity to submit a revised draft of the manuscript. We appreciate the time and effort that you dedicated to providing feedback on our manuscript and are grateful for the insightful comments on and valuable improvements to our paper. We have incorporated most of your suggestions. Those changes are highlighted within the manuscript. Please see below, in red, for a point-by-point response to your comments and concerns.
Line 88 to 95, the study is well justified, you write about 3 groups but you already did not define them, maybe it is better to reserve that for material and methods, because it is not very relevant for the introduction.
Answer: Thank you for your suggestion, we agree. We added a parenthesis to line 95 which refers to chapter 2.2 Participants.
In Materials and methods, we know in which kind of institutions the subjects were recruited, but we do not know in witch condition they responded to the questionnaires (online? Other?), please specify.
Answer: Thank you for pointing this out. All participants answered the questionnaires by hand with the discreet presence of the researchers who explained to them how to complete them. Each participant, when completing the questionnaire, left it at a predetermined point and after some time the researcher went and collected all the questionnaires. The researchers always went to the pre-determined points at the same times and the questionnaires for the exercisers were answered before they started their training.
Table 1, please correct the 176 for Height in OBG
Answer: Thank you. We have made the correction according to your suggestion.

Round 2
Reviewer 2 Report
Comments and Suggestions for Authors
After a major revision, the quanlity of this submittions has been improved greatly. However, I noticed that the marker color of the OBG group in Figure 1 is inconsistent with the color of the histogram.
Author Response
Comment 1. After a major revision, the quanlity of this submittions has been improved greatly. However, I noticed that the marker color of the OBG group in Figure 1 is inconsistent with the color of the histogram.
Responce to comment 1. Thank you for your observation. We changed the color of the OBG group so that there is no inconsistency with the color of the histogram.

Reviewer 3 Report
Comments and Suggestions for Authors
Abstract and Introduction
The abstract section can be improved by summarizing the key findings more concisely. Also, I suggest briefly mentioning the practical implications of the findings to capture the interest of practitioners and academics.
The comprehensive literature assessment includes studies on obesity, consumption habits and the Mediterranean diet. It may, however, benefit from a more succinct synopsis of the major research gaps that this work seeks to fill. In the introduction section, authors should discuss the novelty and contributions of the study. What are the key gaps and questions? Clear research gaps should be discussed. Discuss how the study fills the gap, revealing the uniqueness of the article.
The hypothesis should be clearly defined by strengthening the development of the hypothesis by providing a more detailed rationale based on previous research findings.
Methodology and Results
It would be useful to discuss potential biases introduced by the study design and how they were mitigated. Please report the non-response bias and multicollinearity. Similarly, the data-gathering process is quite long. Please justify this.
I suggest considering a brief justification for selecting each instrument, highlighting their relevance and reliability. Also, it would be good to provide more detailed demographic information to contextualize the sample better.
Discussion and Conclusion
The author (s) should discuss the potential mechanisms underlying the observed relationships in more detail. For example, why might the Mediterranean diet impact body image differently in various BMI groups?
A more nuanced comparison with previous studies, highlighting similarities and differences and speculating on reasons for any discrepancies should be provided. Also, discussing how the findings contribute to or challenge existing theories on body image, eating disorders, and lifestyle interventions would improve the contribution of the paper.
Similarly, theoretical and managerial implications need to be improved. More specific suggestions for managerial implications and theoretical directions are a must. For example, please provide specific recommendations for public health campaigns and policies based on the study's findings. For example, suggest targeted interventions for different BMI groups. Outlining how clinicians can use these findings to support patients with obesity-related body image issues and eating disorders would be useful.
Author Response
Answers to Reviewer 3
Comment 1. The abstract section can be improved by summarizing the key findings more concisely. Also, I suggest briefly mentioning the practical implications of the findings to capture the interest of practitioners and academics.
Answer to Comment 1. Thank you for your suggestion. Following a detailed examination of the comments you have provided; we have amended the abstract in accordance with your recommendations.
Comment 2. The comprehensive literature assessment includes studies on obesity, consumption habits and the Mediterranean diet. It may, however, benefit from a more succinct synopsis of the major research gaps that this work seeks to fill. In the introduction section, authors should discuss the novelty and contributions of the study. What are the key gaps and questions? Clear research gaps should be discussed. Discuss how the study fills the gap, revealing the uniqueness of the article.
Answer to Comment 2. We agree with your suggestion. We have reformatted the introduction and changed the order of the variables as indicated in the title of the article. In the last paragraph we have added the gaps in the literature and the questions they raise (lines 49-65 and 87-100).
Comment 3. The hypothesis should be clearly defined by strengthening the development of the hypothesis by providing a more detailed rationale based on previous research findings.
Answer to Comment 3. We agree with your suggestion. We have reformatted the hypothesis (lines 98-100).
Comment 4. It would be useful to discuss potential biases introduced by the study design and how they were mitigated. Please report the non-response bias and multicollinearity. Similarly, the data-gathering process is quite long. Please justify this.
Answer to Comment 4. Thank you for your suggestions. I will start with the last one about the length of the process. As you are well aware, conducting surveys with questionnaires hand-delivered to such a large sample is a time-consuming process. All researchers teach at universities and the time available for research is limited. Also, it was not possible to find an infinite number of people willing and able to participate in the survey, so we had to be patient until the required number of the sample was filled. These are common difficulties in research in Greece. The rest is answered meticulously in the section "Limitations" (lines 491-508).
Comment 5. I suggest considering a brief justification for selecting each instrument, highlighting their relevance and reliability. Also, it would be good to provide more detailed demographic information to contextualize the sample better.
Answer to Comment 5. Thank you for your suggestions. The questionnaires used in the survey are world renowned and have been used in thousands of surveys. They are documented for both validity and reliability. You can read about the questionnaires in chapter 2.3 Instrumentation (lines 140 - 215). You can read about reliability in chapter 3.1.3 Reliability analysis (lines 264 - 272).
Comment 6. The author (s) should discuss the potential mechanisms underlying the observed relationships in more detail. For example, why might the Mediterranean diet impact body image differently in various BMI groups?
Answer to Comment 6. Thank you for your suggestions. In the example you suggest, what we need to observe is that the Mediterranean diet does not have a different effect depending on the different BMIs of the groups. The answer is that the Mediterranean diet has a positive effect on all those who follow it faithfully. It's just that the OBG group had a moderate adherence to the Mediterranean diet, whereas the NBG group had a high adherence. Therefore, the differences were not due to BMI, but to the groups' adherence to the Mediterranean diet. In conclusion, it seems that people with a high adherence to the Mediterranean diet and regular leisure-time physical activity have benefits for both their physical and mental health.
Comment 7. A more nuanced comparison with previous studies, highlighting similarities and differences and speculating on reasons for any discrepancies should be provided. Also, discussing how the findings contribute to or challenge existing theories on body image, eating disorders, and lifestyle interventions would improve the contribution of the paper.
Answer to Comment 7. Thank you for your suggestions. We believe that both in the introduction and in the discussion we have listed all the research that is relevant to our topic. No research was found that had a similar topic to recreational physical activity and the Mediterranean diet. As we mentioned in the first round, we are interested in where we are as Mediterranean people in relation to our return to the Mediterranean lifestyle after the economic crisis of 2009-2016, which pushed us towards a diet of convenience and packaged products. In the discussion section we list the main proposals for improving the Mediterranean lifestyle (lines 483 - 488).
Comment 8. Similarly, theoretical and managerial implications need to be improved. More specific suggestions for managerial implications and theoretical directions are a must. For example, please provide specific recommendations for public health campaigns and policies based on the study's findings. For example, suggest targeted interventions for different BMI groups. Outlining how clinicians can use these findings to support patients with obesity-related body image issues and eating disorders would be useful.
Answer to Comment 8. Thank you for your suggestions. In response to your suggestions above, we refer you to Chapter 4.3 Implications (lines 511 - 533). This chapter provides detailed suggestions for each of your recommendations. Finally, our recommendations are addressed not only to clinicians, but also to nutritionists and exercise specialists, who should work together for better and more scientific results.

Reviewer 4 Report
Comments and Suggestions for Authors
I do not have comment.
Thank you for revision efforts.
Author Response
Thank you and we appreciate your effort to help improve the article.

Round 3
Reviewer 3 Report
Comments and Suggestions for Authors
Thank you for the revisions and responses. I hope my suggestions have contributed to the improvement of the study.